# MultiHop-RAG: Benchmarking Retrieval-Augmented Generation for Multi-Hop Queries

**Yixuan Tang , Yi Yang**
The Hong Kong University of Science and Technology
{yixuantang,imyiyang}@ust.hk

## Abstract

Retrieval-augmented generation (RAG) augments large language models (LLM) by retrieving relevant knowledge, showing promising potential in mitigating LLM hallucinations and enhancing response quality, thereby facilitating the great adoption of LLMs in practice. However, we find that existing RAG systems are inadequate in answering multi-hop queries, which require retrieving and reasoning over multiple pieces of supporting evidence. Furthermore, to our knowledge, no existing RAG benchmarking dataset focuses on multi-hop queries. In this paper, we develop a novel dataset, **MultiHop-RAG**, which consists of a knowledge base, a large collection of multi-hop queries, their ground-truth answers, and the associated supporting evidence. We detail the procedure of building the dataset, utilizing an English news article dataset as the underlying RAG knowledge base. We demonstrate the benchmarking utility of MultiHop-RAG in two experiments. The first experiment compares different embedding models for retrieving evidence for multi-hop queries. In the second experiment, we examine the capabilities of various state-of-the-art LLMs, including GPT-4, PaLM, and Llama2-70B, in reasoning and answering multi-hop queries given the evidence. Both experiments reveal that existing RAG methods perform unsatisfactorily in retrieving and answering multi-hop queries. We hope MultiHop-RAG will be a valuable resource for the community in developing effective RAG systems, thereby facilitating greater adoption of LLMs in practice. We make the dataset and benchmarking code publicly available via GitHub[1].

## 1 Introduction

The emergence of large language models (LLMs), such as ChatGPT, has fostered a wide range of innovations, powering intelligent chatbots and other natural language processing (NLP) applications (OpenAI, 2023). One promising use case is Retrieval-Augmented Generation (RAG) (Asai et al., 2023), which optimizes the output of a large language model by referencing an external knowledge base outside of the LLM training data sources before generating a response. RAG improves LLM's response (Borgeaud et al., 2022) and also mitigates the occurrence of hallucinations, thereby enhancing the models' credibility (Gao et al., 2023). LLM-based frameworks, such as LlamaIndex (Liu, 2022) and LangChain (Chase, 2022), specialize in supporting RAG pipelines.

In real-world Retrieval-Augmented Generation (RAG) applications, a user's query often necessitates retrieving and reasoning over evidence from multiple documents, a process known as **multi-hop query**. For instance, consider financial analysis using a database of financial reports. A financial analyst might query, *Which company among Google, Apple, and Nvidia reported the largest profit margins in their third-quarter reports for 2023?* or inquire about a specific company's performance over time, such as *How does Apple's sales trend look over the past three years?* These queries require evidence from multiple documents to formulate an answer. Due to the multifaceted nature of such queries, involving information from various

---

[1]https://github.com/yixuantt/MultiHop-RAG

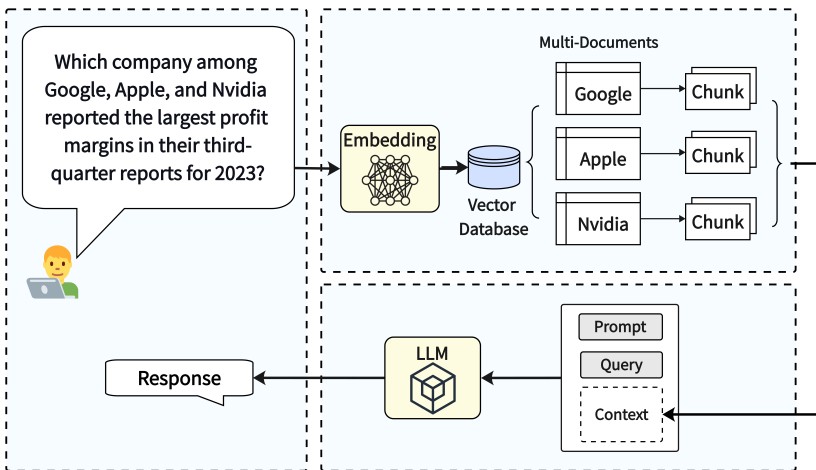

Figure 1: RAG with multi-hop query.

sources, traditional similarity matching methods like cosine similarity between query and financial report chunk embeddings might not yield optimal results. We demonstrate this multi-hop retrieval process in Figure 1.

However, existing RAG benchmarks, such as RGB (Chen et al., 2023) and RECALL (Liu et al., 2023), mainly evaluate a simple case where the answer of a query can be retrieved and solved using one single piece of evidence. None of these benchmarks assess the retrieval and reasoning capability of LLMs for complex multi-hop queries. To address this gap and make RAG benchmarking more closely resemble real-world scenarios, in this paper, we introduce **MultiHop-RAG**. To our knowledge, MultiHop-RAG is one of the first RAG datasets focusing specifically on multi-hop queries.

Based on the RAG queries commonly encountered in real-world scenarios, we first categorize multi-hop queries into four types: *Inference query*, *Comparison query*, *Temporal query*, and *Null query*. The first three types — Inference, Comparison, and Temporal — require the retrieval and analysis of evidence from multiple sources, encompassing tasks like inferring relationships, comparing data points, and sequencing events over time. The Null query represents a scenario where the query cannot be derived from the knowledge base. This category is crucial for assessing whether an LLM might hallucinate an answer to a multi-hop query when the retrieved text lacks relevance.

We construct our RAG knowledge base using a collection of news articles. Using GPT-4 as a data generator, we then take an extensive procedure to construct a diverse set of multi-hop queries, each requiring the retrieval and reasoning over multiple documents. An example of query construction is shown in Table 1. First, we begin by extracting factual sentences from each news article as evidence. For example, an extracted piece of evidence from an article may state: "Back then, just like today, home prices had boomed for years before Fed officials were ultimately forced to hike interest rates aggressively in an attempt to fight inflation." Second, we input each evidence piece into GPT-4, prompting it to rephrase the evidence into a claim. This claim is clarified with a disambiguated topic and entity. For instance, GPT-4 might rephrase the aforementioned evidence into: "Federal Reserve officials were forced to aggressively hike interest rates to combat inflation after years of booming home prices", identifying "Interest rate hikes to combat inflation" as the topic and "Federal Reserve" as the entity. These topics and entities act as bridges for constructing multi-hop queries, known as bridge-topic or bridge-entity. Next, we use GPT-4 to generate specific multi-hop queries related to the same bridge-topic or bridge-entity, accompanied by the correct answers. Lastly, we undertake a validation step to ensure the data quality.

We demonstrate the benchmarking capabilities of MultiHop-RAG using two experiments, utilizing a RAG system implemented with LlamaIndex (Liu, 2022). The first experiment

| News source | Fortune Magazine | The Sydney Morning Herald |
|---|---|---|
| Evidence | Back then, just like today, home prices had boomed for years before Fed officials were ultimately forced to hike interest rates aggressively in an attempt to fight inflation. | Postponements of such reports could complicate things for the Fed, which has insisted it will make upcoming decisions on interest rates based on what incoming data say about the economy. |
| Claim | Federal Reserve officials were forced to aggressively hike interest rates to combat inflation after years of booming home prices. | The Federal Reserve has insisted that it will base its upcoming decisions on interest rates on the incoming economic data. |
| Bridge-Topic | Interest rate hikes to combat inflation | Interest rate decisions based on economic data |
| Bridge-Entity | Federal Reserve | Federal Reserve |
| Query | Does the article from Fortune suggest that the Federal Reserve's interest rate hikes are a response to past conditions, such as booming home prices, while The Sydney Morning Herald article indicates that the Federal Reserve's future interest rate decisions will be based on incoming economic data? | |
| Answer | Yes | |

Table 1: An example of a multi-hop query, including supporting evidence from two news articles, the paraphrased claim, the bridge-topic and bridge-entity, and the corresponding answer.

involves a comparison of different embedding models for retrieving relevant evidence for multi-hop queries. In the second experiment, we assess the reasoning and answering abilities of various state-of-the-art LLMs, including GPT-4, GPT-3.5, PaLM, Claude-2, Llama2-70B, and Mixtral-8x7B, for multi-hop queries when retrieved text is provided. The results from both experiments indicate that the current RAG implementations are inadequate for effectively retrieving and answering multi-hop queries. We publicly release this challenging MultiHop-RAG dataset and hope it will be a valuable resource for the community in developing and benchmarking RAG systems, thereby unleashing the great potential of generative AI in practice.

## 2 RAG with multi-Hop queries

### 2.1 Retrieval-augmented Generation (RAG)

In an RAG application, we utilize an external corpus, denoted as $\mathcal{D}$, which comprises multiple documents and serves as the knowledge base. Each document within this corpus, represented as $d_i \in \mathcal{D}$, is segmented into a set of chunks. These chunks are then transformed into vector representations using an embedding model and stored in an embedding database. Given a user query $q$, the system typically retrieves the top-K chunks that best match the query. These chunks constitute the **retrieval set** for query $q$, represented as $\mathcal{R}_q = \{r_1, r_2, ..., r_K\}$. The retrieved chunks, combined with the query and an optional prompt, are then fed into an LLM to generate a final answer, following the format: $\text{LLM}(q, \mathcal{R}_q, \texttt{prompt}) \rightarrow$ answer.

### 2.2 Multi-Hop Query

We define a multi-hop query as one that requires *retrieving* and *reasoning* over multiple pieces of supporting evidence to provide an answer. In other words, for a multi-hop query $q$, the chunks in the retrieval set $\mathcal{R}_q$ collectively provide an answer to $q$. For example, the query "Which company among Google, Apple, and Nvidia reported the largest profit margins in their third-quarter reports for 2023?" requires 1) retrieving relevant pieces of evidence related to profit margins from the reports of the three companies; 2) generating an answer by comparing and reasoning from the multiple pieces of retrieved evidence. This differs from a single-hop query such as "What is Google's profit margin in the third-quarter reports for 2023," where the answer can be directly derived from a single piece of evidence.

Based on the queries commonly used in real-world RAG systems, we identify four types of multi-hop queries. For each type, we present a hypothetical query within the context of a financial RAG system, where the knowledge base consists of a collection of annual reports.

**Inference query:** For such a query $q$, the answer is deduced through reasoning from the retrieval set $\mathcal{R}_q$. An example of an inference query might be: *Which report discusses the supply chain risk of Apple, the 2019 annual report or the 2020 annual report?*

**Comparison query:** For such a query $q$, the answer requires a comparison of evidence within the retrieval set $\mathcal{R}_q$. For instance, a comparison query might ask: *Did Netflix or Google report higher revenue for the year 2023?"*

**Temporal query:** For such a query $q$, the answer requires an analysis of the temporal information of the retrieved chunks. For example, a temporal query may ask: *Did Apple introduce the AirTag tracking device before or after the launch of the 5th generation iPad Pro?*

**Null query:** For such as query $q$, the answer cannot be derived from the retrieved set $\mathcal{R}_q$. We include the null query to assess the generation quality, especially regarding the issue of hallucination. For a null query, even though a retrieved set is provided, an LLM should produce a null response instead of hallucinating an answer. For example, assuming ABCD is a non-existent company, a null query might ask: *What are the sales of company ABCD as reported in its 2022 and 2023 annual reports?*

### 2.3 Evaluation Metrics

An RAG system handling multi-hop queries can be assessed from two key aspects: retrieval evaluation and generation evaluation.

**Retrieval Evaluation:** Evidently, the quality of the retrieval set $\mathcal{R}_q$ determines the final generation quality. We compare the retrieved set with the ground truth evidence associated with each query, except for the null queries, as they have no evidence to derive from. Assuming the top-K chunks are retrieved, i.e., $|\mathcal{R}_q| = K$, we use retrieval evaluation metrics including Mean Average Precision at K (MAP@K), Mean Reciprocal Rank at K (MRR@K), and Hit Rate at K (Hit@K). *MAP@K* measures the average top-K retrieval precision across all queries. *MRR@K* calculates the average of the reciprocal ranks of the first relevant chunk for each query, considering the top-K retrieved set. *Hit@K* metric measures the fraction of evidence that appears in the top-K retrieved set.

**Response Evaluation:** Since the multi-hop query requires reasoning over multiple pieces of retrieved chunks, we can also evaluate the reasoning capability of the LLM by comparing the LLM response with the ground truth answer of the query.

## 3 A Benchmarking Dataset: MultiHop-RAG

In this section, we provide detailed information on the construction of the MultiHop-RAG dataset. Specifically, we describe the process of creating a set of multi-hop queries, along with the corresponding ground truth evidence sets and answers derived from a collection of news articles.

### 3.1 MultiHop-RAG Construction

**Step 1: Dataset Collection.** We download a news dataset using the mediastack API [2], a REST API interface delivering worldwide news data. The news data source comprises various English-language websites covering a range of news categories: entertainment, business, sports, technology, health, and science. To mimic real-world RAG scenarios, where the knowledge base data, such as an enterprise's internal data, may differ from the LLMs' training data, we select news articles published from September 26, 2023, to December 26, 2023. This timeframe extends beyond the knowledge cutoff of some widely-used LLMs, including ChatGPT and LLaMA, as of the time of writing. This selection also helps in teasing

---
[2]https://mediastack.com/

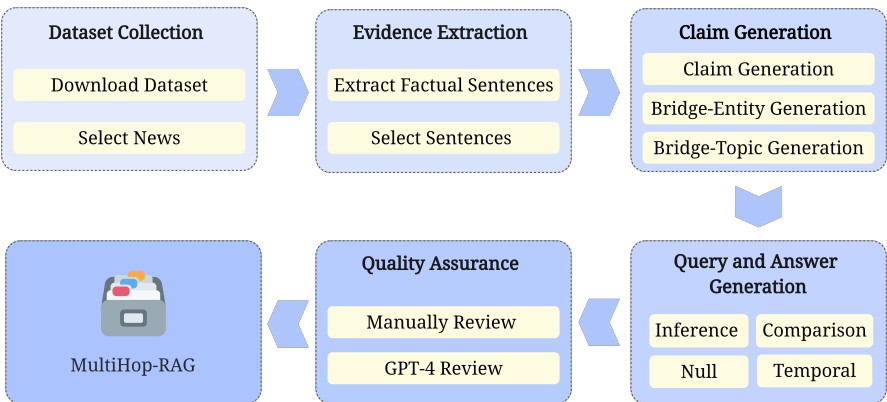

Figure 2: MultiHop-RAG Construction Pipeline.

out the possibility of the underlying LLM having been exposed to these news articles. We only keep articles with a token length greater than or equal to 1,024. Every news article is paired with metadata, including the title, publish date, author, category, URL, and news source.

**Step 2: Evidence Extraction.** For each article, we extract factual or opinion sentences using a trained language model [3]. These factual sentences are later used as evidence for answering multi-hop queries. We retain only those news articles containing evidence that may have overlapping keywords with other news articles. This allows us to later create multi-hop queries where the answer's evidences are drawn from multiple sources.

**Step 3: Claim, Bridge-Entity, Bridge-Topic Generation.** Our goal is to use GPT-4 to automatically generate high-quality multi-hop queries using the evidence set. However, the raw evidence obtained from Step 2 is not ideal for query generation due to inconsistency in linguistic structure. For example, some pieces of evidence use pronouns to refer to subjects and lack the actual entity in the text. To address this, we employ GPT-4 to paraphrase the evidence, which we refer to as *claims*, given the original evidence and its context. To ensure consistency between the generated claim and the evidence, we further perform fact-checking using the UniEval (Zhong et al., 2022) framework to verify the alignment between the evidence and claim. Appendix A presents the prompt used for GPT-4 for claim generation.

**Bridge-Entity and Bridge-Topic:** The shared entity or topic across pieces of evidence is referred to as the bridge-entity or bridge-topic. These bridge-entities or bridge-topics can be used to link different pieces of evidence from which a multi-hop query's answer is derived. For example, in a claim such as *"Google reports its third-quarter results for 2023, showcasing a detailed overview of its financial performance, including revenue growth, profit margins"*, the term *profit margin* can be viewed as a bridge-topic and the term *Google* can be viewed as a bridge-entity that links the different pieces of evidence. We prompt GPT-4 to identify the bridge-entity and bridge-topic for each claim. Appendix A also presents the prompt used for GPT-4 for bridge generation.

**Step 4: Query and Answer Generation.** In this step, we leverage the bridge-entity or bridge-topic to generate multi-hop queries. Specifically, we first group the claims having the same bridge-entity or bridge-topic into a claim set. We restrict the claim set to have at least two claims but no more than four claims. For each type of query, we feed the claim set to GPT-4 and prompt it with an instruction to generate a query with information from each claim. Below, we explain the specifications for different multi-hop query types. In the construction of each query, we also include the source of the news article where the

---

[3]https://huggingface.co/lighteternal/fact-or-opinion-xlmr-el

| Num. of Evidence Needed | Count | Percentage |
|---|---|---|
| 0 (Null Query) | 301 | 11.78% |
| 2 | 1078 | 42.18% |
| 3 | 779 | 30.48% |
| 4 | 398 | 15.56% |
| Total | 2,556 | 100.00 % |

Table 2: The distribution of the number of evidence required to answer multi-hop queries in MultiHop-RAG.

| Query Category | Entry Count | Percentage |
|---|---|---|
| Inference Query | 816 | 31.92% |
| Comparison Query | 856 | 33.49% |
| Temporal Query | 583 | 22.81% |
| Null Query | 301 | 11.78% |
| Total | 2,556 | 100.00 % |

Table 3: The distribution of query types in MultiHop-RAG.

supporting evidence is associated with mimicking real-world RAG scenarios. Appendix A presents the prompts used for GPT-4 for query generation.

**Inference Query:** These queries are formulated by synthesizing the various characterizations of the bridge-entity across multiple claims, with the final answer being the identification of the entity itself.

**Comparison Query:** These queries are structured to compare the similarities and differences related to the bridge entity or topic. The resultant answer to such queries is typically a definitive "yes" or "no", based on the comparison.

**Temporal Query:** These queries explore the temporal ordering of events across different points in time. The answer to such queries is typically a "yes" or "no" or a single temporal indicator word like "before" or "after".

**Null Query:** Null query is a query whose answer cannot be derived from the retrieved set. To create null queries, we generate multi-hop queries using entities that do not exist in the existing bridge-entities. To add complexity, we also include fictional news source metadata when formulating these questions, ensuring that the questions do not reference any contextually relevant content from the knowledge base. The answer to the null query should be "insufficient information" or similar.

**Step 5: Quality Assurance.** Finally, we use two approaches to reassure the dataset quality. First, we manually review a subset sample of the generated multi-hop queries, their corresponding evidence sets, and the final answers. The results of the manual examination indicate a high degree of accuracy and data quality. Second, we utilize GPT-4 to assess each example in the dataset against the following criteria: 1) The generated query must utilize all provided evidence in formulating the response; 2) The query should be answerable solely based on the provided evidence; 3) The response to the generated query should be either a single word or a specific entity; 4) The query must conform to its designated query type.

## 3.2 Descriptive Statistics

The MultiHop-RAG dataset contains six different types of news articles, covering 609 distinct news, with an average of 2,046 tokens. The distribution of the news categories is shown in Table 4. MultiHop-RAG contains four types of multi-hop queries and the distribution of these queries is shown in Table 3. In total, about 88% of queries in the dataset are non-null queries where answers can be retrieved and reasoned from the knowledge base. In addition, the form of queries exhibits considerable diversity. Approximately 27% of interrogative queries start with "does," around 15% initiate with "what," a similar proportion start "which," and 14% begin with "who," with the remainder incorporating a small percentage of other interrogative words such as "when." Moreover, the number of evidence required to answer a multi-hop query varies. Table 2 shows the distribution of evidence numbers for each query in the dataset. Around 42% of queries can be answered using two pieces of evidence, while approximately 30% and 15% of queries can be answered using three or four pieces of evidence, respectively.

| Category | Avg. Tokens | Entry Count |
|---|---|---|
| technology | 2262.3 | 172 |
| entertainment | 2084.3 | 114 |
| sports | 2030.6 | 211 |
| science | 1745.5 | 21 |
| business | 1723.8 | 81 |
| health | 1481.1 | 10 |
| total | 2046.5 | 609 |

Table 4: Descriptive statistics of the news article knowledge base in MultiHop-RAG.

## 4 Benchmarking RAG system using MultiHop-RAG

MultiHop-RAG can be used as a benchmark for various RAG-related tasks. Broadly speaking, RAG-related tasks can be categorized as *retrieval-related tasks* and *generation-related tasks*. A retrieval-related task focuses on retrieving relevant text from the knowledge base, while a generation-related task focuses on generating high-quality responses given the retrieved text. In this section, we showcase two use cases for each task where MultiHop-RAG can be employed.

### 4.1 Retrieval-related Task

An important design choice in an RAG system is the selection of the embedding model. An embedding model converts data into numerical vectors and subsequently stores these vectors in embedding databases. In this experiment, we evaluate different embedding models by examining their retrieval quality.

**Experiment Setup:** We implement an RAG system using the LlamaIndex framework (Liu, 2022). We partition the documents in the MultiHop-RAG knowledge base into chunks, each consisting of 256 tokens. We then convert the chunks using an embedding model and save the embeddings into a vector database. Similarly, in the retrieval step, we convert a query using the same embedding model and retrieve the top-K most relevant chunks that have the highest cosine similarity with the query embedding. In this experiment, we test a variety set of embedding models, including the ada-embeddings by OpenAI (text-embedding-ada-002, text-search-ada-query-001), voyage-02 [4], llm-embedder (Zhang et al., 2023), bge-large-en-v1.5 (Xiao et al., 2023), jina-embeddings-v2-base-en (Günther et al., 2023), e5-base-v2 (Wang et al., 2022), and instructor-large (Su et al., 2023). NULL queries are excluded in this experiment because there is no matching evidence to the query. Additionally, we also include a Reranker module to examine the retrieval performance, using bge-reranker-large (Xiao et al., 2023). After retrieving 20 related chunks using the embedding model, we further select the top-K chunks using the Reranker.

**Experiment Result:** Table 5 shows the retrieval result of using different embedding models. It shows that there is still a significant gap in retrieving relevant evidence for the multi-hop queries. While Rerank can effectively improve retrieval relevance, the highest Hits@10 is only 0.7467 when the Reranker technique is used. Moreover, the drop in the highest Hits@4 to 0.6625 is worrisome. In practical RAG systems, the underlying LLM often has a context window limit. As a result, the number of retrieved chunks is usually restricted to a small number. The low values of the retrieval metrics highlight the challenges in retrieving relevant pieces of evidence for multi-hop queries when using direct similarity matching between the multi-hop query and text chunks.

### 4.2 Generation-related Task

The underlying LLMs play a crucial role in generating responses in an RAG system. In this experiment, we evaluate the quality of generated responses under two different settings. In

---

[4]https://www.voyageai.com/

| Embedding | Without Reranker | | | | With bge-reranker-large | | | |
|---|---|---|---|---|---|---|---|---|
| | MRR@10 | MAP@10 | Hits@10 | Hits@4 | MRR@10 | MAP@10 | Hits@10 | Hits@4 |
| text-embedding-ada-002 | 0.4203 | **0.3431** | 0.6381 | 0.5040 | 0.5477 | 0.4625 | 0.7059 | 0.6169 |
| text-search-ada-query-001 | 0.4203 | **0.3431** | 0.6399 | 0.5031 | 0.5483 | 0.4625 | 0.7064 | 0.6174 |
| llm-embedder | 0.2558 | 0.1725 | 0.4499 | 0.3189 | 0.4250 | 0.3059 | 0.5478 | 0.4756 |
| bge-large-en-v1.5 | **0.4298** | 0.3423 | **0.6718** | **0.5221** | 0.5630 | 0.4759 | 0.7183 | 0.6364 |
| jina-embeddings-v2-base-en | 0.0621 | 0.0310 | 0.1479 | 0.0802 | 0.1412 | 0.0772 | 0.1909 | 0.1639 |
| intfloat/e5-base-v2 | 0.1843 | 0.1161 | 0.3556 | 0.2334 | 0.3237 | 0.2165 | 0.4176 | 0.3716 |
| voyage-02 | 0.3934 | 0.3143 | 0.6506 | 0.4619 | **0.5860** | **0.4795** | **0.7467** | **0.6625** |
| hkunlp/instructor-large | 0.3458 | 0.2650 | 0.5717 | 0.4229 | 0.5115 | 0.4118 | 0.6590 | 0.5775 |

Table 5: Retrieval performance of different embedding models.

| Models | Accuracy | |
|---|---|---|
| | Retrieved Chunk | Ground-truth Chunk |
| GPT-4 | **0.56** | **0.89** |
| ChatGPT | 0.44 | 0.57 |
| Llama-2-70b-chat-hf | 0.28 | 0.32 |
| Mixtral-8x7B-Instruct | 0.32 | 0.36 |
| Claude-2.1 | 0.52 | 0.56 |
| Google-PaLM | 0.47 | 0.74 |

Table 6: Generation accuracy of LLMs.

the first setting, we employ the best-performing retrieval model, namely voyage-02 with bge-reranker-large, as indicated in Table 5, to retrieve the top-K texts and then feed them into the LLM. In the second setting, we use the ground-truth evidence associated with each query as the retrieved text for the LLM. This setting represents a ceiling performance for testing the LLM's response capabilities, as it utilizes the actual evidences.

**Experiment Setup:** In the first experiment, we retrieve top-6 chunks so that the total length of the retrieved text does not exceed 2,048. All queries in MultiHop-RAG are tested in the experiment. In the second experiment, since the null queries do not have associated evidence, we exclude this type of query in the experiment. For the LLMs used in the experiment, we consider state-of-the-art commercial models, including GPT-4 (OpenAI, 2023), GPT-3.5, Claude-2 (Anthropic, 2023), and Google-PaLM (Google, 2023). We obtain answers using the provided API of the respective models. We also assess some open-source models, including Mixtral-8x7b-instruct (Jiang et al., 2024) and Llama-2-70b-chat-hf (Touvron et al., 2023).

**Experiment Results:** Table 6 shows the response accuracy of different LLMs. First, we can see that the response accuracy rate using the retrieved chunks is not satisfactory, with the state-of-the-art GPT-4 model achieving only 0.56 accuracy. This is expected because the retrieval component falls short in retrieving relevant evidence from the knowledge base. Second, even when we provide the LLM with ground-truth evidence, we can see that the response accuracy is far from being perfect. Open source LLM, such as Llama02-70B and Mixtral-8x7B, only achieve an accuracy of 0.32 and 0.36, respectively. GPT-4 achieves strong reasoning capability with an accuracy of 0.89, followed by the second-based LLM Google-PaLM with an accuracy of 0.74.

Figure 3 shows the detailed results of different query types for GPT-4 and Mixtral-8x7B-instruct. Both models show relatively high robustness on null queries, meaning they are generally good at determining when a query cannot be answered based on the retrieved text. This is encouraging because one benefit of RAG is to mitigate the LLM hallucination issue by augmenting LLM with retrieval knowledge. However, Mixtral-8x7B model performs significantly worse than the GPT-4 in comparison and temporal queries. Upon reviewing the incorrect responses, we find that Mixtral-8x7B fails to accurately handle logical negation, leading to misinterpretation of statements and thus a low performance in the comparison queries. In addition, Mixtral-8x7B often fails to correctly identify the chronological order of events, which is crucial for answering temporal queries where timing is a key factor. Taken together, this experiment demonstrates that there is still room for improvement in

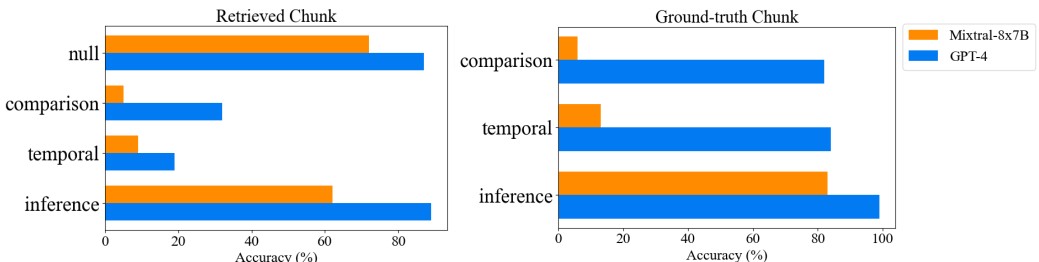

Figure 3: Generation accuracy for different query types.

the reasoning capabilities of LLMs, particularly those that are open-source, for multi-hop queries.

### 4.3 Other Use Cases

Beyond embedding models and LLM generation, there are other areas worth exploring. For example, query decomposition is a widely utilized technique in RAG frameworks, such as LLamaIndex. This process involves breaking down the query into sub-questions; it targets a single document for retrieval and integrates the information subsequently, thereby potentially enhancing retrieval accuracy. Another advanced and promising approach involves building LLM-based agents that can automatically plan and execute multi-hop queries, such as AutoGPT (Gravitas, 2023). AutoGPT breaks down complex instructions into smaller steps. It analyzes feedback to continuously refine its response until the final decision. Another promising direction is the exploration of hybrid retrieval approaches. These methods combine the precision of keyword-based search with the contextual understanding afforded by embedding. By leveraging the strengths of both approaches, hybrid models aim to deliver more accurate and relevant search results. Overall, we believe that there are many potential areas for enhancing RAG's performance on multi-hop queries, and the curated dataset MultiHop-RAG can be a valuable resource to the community.

## 5    Related Work

**RAG Evaluation:** As RAG systems gain increasing popularity, a variety of RAG benchmarking datasets and evaluation tools have been developed. For instance, RGB (Chen et al., 2023) and RECALL (Liu et al., 2023) evaluate the performance of LLMs in generating responses for RAG systems under conditions involving noisy, integrative, and counterfactual queries. However, both datasets primarily focus on evaluating the generation aspect of RAG systems without specifically addressing their retrieval accuracy. In addition, recent advancements have been made in automated RAG evaluation tools, such as ARES (Saad-Falcon et al., 2023) and RAGAS (Es et al., 2023). These tools utilize LLMs to automatically assess the quality of RAG generation, yet they do not introduce benchmarking datasets. Our work introduces one of the first RAG benchmarking datasets, consisting of a knowledge base, a large collection of multi-hop queries, their ground-truth answers, and the associated supporting evidence, thereby complementing existing RAG evaluations.

**Retrieval datasets:** Apart from the context of RAG, several benchmarking datasets exist for information retrieval evaluation. The FEVER (Fact Extraction and VERification) dataset, for instance, contains claims classified as Supported, Refuted, or NotEnoughInfo by the given Wikipedia article (Thorne et al., 2018). Similarly, the SciFact dataset comprises scientific claims paired with evidence-containing abstracts (Wadden et al., 2020). However, the claims in both datasets are single-hop statements, and the supporting evidence is from one single article, in contrast to the multi-hop queries discussed in this paper. Another dataset, HoVer, involves claims that require extracting and reasoning from multiple Wikipedia articles (Jiang et al., 2020). However, unlike our dataset, HoVer focuses solely on classifying claims as

either supported or not supported by the articles without evaluating an LLM generation step. Moreover, in HoVer, the Wikipedia articles from which evidence is drawn are given for claim verification, which is significantly different from our setting, where relevant pieces of evidence need to be extracted from a large knowledge base. Separately, (Kamalloo et al., 2023) evaluates a range of commercial embedding APIs for information retrieval, but this evaluation is not contextualized within the framework of RAG systems either.

**Multi-document QA datasets:** Question-answering (QA) is a fundamental task in NLP, and several popular benchmarks, such as HotpotQA (Yang et al., 2018), MultiRC (Khashabi et al., 2018), and 2WikiMultiHopQA (Ho et al., 2020), aim to answer the question with multiple sources. This task is similar to our multi-hop query RAG task, as both involve reasoning from multiple sources of information. However, existing datasets are not tailored for the RAG task as they lack a retrieval step. Even modifying a multi-hop query dataset to assess retrieval performance by integrating the input evidence into a corpus does not adequately reflect the demands of real-world RAG tasks. A primary concern is that these datasets, often sourced from Wikipedia, overlap with training data used for large language models (LLMs). In practical RAG scenarios, datasets typically comprise private corpora with metadata like titles or timestamps. This overlap can compromise the evaluation process, raising concerns that LLM responses may depend more on memorized training data rather than on reasoning from the retrieved knowledge base. The proposed dataset, in contrast, is specifically crafted for RAG tasks, including those designed to simulate real-life scenarios. It addresses these issues by introducing a very recent corpus paired with relevant metadata. This dataset can serve as a benchmark to evaluate multi-evidence retrieval and multi-hop evidence generation capabilities.

# 6  Conclusion

In this work, we introduce MultiHop-RAG, a novel and unique dataset designed for queries that require retrieval and reasoning from multiple pieces of supporting evidence. These types of multi-hop queries represent user queries commonly encountered in real-world scenarios. MultiHop-RAG consists of a knowledge base, a large collection of multi-hop queries, their ground-truth answers, and the associated supporting evidence. This paper details the creation process of MultiHop-RAG, employing a hybrid approach that integrates human effort with GPT-4. Additionally, we explore two use cases of MultiHop-RAG in the benchmarking of RAG systems, thereby highlighting the potential applications of this dataset. By publicly releasing MultiHop-RAG, we aim to provide a valuable resource to the community, contributing to the advancement and benchmarking of RAG systems.

# Limitations

This work has several limitations that can be improved in future research. First, our ground truth answers are restricted to simple responses such as "yes", "no", entity names, or temporal indicators like "before" or "after" to facilitate the use of a straightforward accuracy metric for evaluating generation performance. Future work could consider allowing free text as answers and employing more sophisticated metrics to assess generation quality. Second, the current dataset limits supporting evidence for a query to a maximum of four pieces. Future work can extend the dataset by including queries that require retrieving and reasoning from even more evidence. Lastly, while our experiments utilize a basic RAG framework using LlamaIndex, future work could involve evaluating the answering of multi-hop queries using more advanced RAG frameworks or LLM-agent frameworks.

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

# A    Appendix A: GPT-4 Prompts Used for Data Generation

We present the prompts used for guiding GPT-4 for data generation. Table 7 shows the prompt used for claim generation, along with the corresponding topics and entities within these claims. Table 8, Table 9, and Table 10 respectively show the prompts used for generating multi-hop queries of the inference, comparison, and temporal types.

---

A "claim" is a statement or assertion made within a text expressing a belief, opinion, or fact. Given evidence from the original context, please extract one claim and its associated topics.

Note: The claim should not contain ambiguous references, such as 'he',' she,' and' it', and should use complete names. If there are multiple topics, give the most dominant one. The target of the claim (one entity)is the specific individual, group, or organization that the statement or assertion within a text is directed towards or about which it is making a case. The topic of the claim should be a simple phrase representing the claim's central argument concept. If there is no claim, please leave it blank. Please generate a claim based on the given evidence. Don't generate the evidence yourself.

Please give the response following this format:
Evidence: [original context]
Claims: [extract claim]
Claim Target: [target]
Claim Topic: [topic]

Here are examples:
⟨examples⟩
Now, it's your turn.
⟨News⟩
⟨evidence⟩

---

Table 7: Claim Generation Prompting

---

A multi-hop question is a query requiring multiple inferential leaps or accessing several pieces of information from different locations or sources to arrive at an answer. The following are news articles' metadata and claims come from the articles. All the claims from the article are related to a similar target. Your task is to generate one multi-hop inference question based on the claims. Here are some instructions:
1. Find the Connection: The connection between claims is ⟨target⟩, which is how these key pieces of information are related or how they can be combined to form a more complex idea.
2. Formulate the Question: Create a question that cannot be answered by relying on just one of the sentences but instead requires understanding and linking the information from all of the sources. The answer is ⟨target⟩.
3. Ensure Coherence: Make sure the question flows logically from the combined information and is clear and unambiguous.
4. Use the keywords: ⟨key set⟩

⟨examples⟩
Context:
⟨Context⟩

---

Table 8: Inference Query Generation Prompting

⟨*Context*⟩

The above are news articles' metadata and claims come from the articles. All the claims from the articles are related to a similar target. Your task is to generate one comparison question based on all the claims from different sources. This question needs to compare some factual elements of the claims that are explicitly stated to find where they agree or differ. The correct answer to this question is expressed as a comparative adjective, a statement of alignment, a simple yes or no. To generate a comparative question from claims, you need to use the following keywords: ⟨*key set*⟩

The Good Comparison Questions:
⟨*examples*⟩
Your Comparison Question:

Table 9: Comparison Query Generation Prompting

⟨*Context*⟩

Please create a time-sensitive comparison question using metadata and excerpts from multiple news articles. That is to compare the consistency or sequence of reports on similar topics at multiple different time points. If it is to compare the consistency, please clearly mention the news source and time in the question using ⟨*time frame*⟩. If it is to compare sequences of reports, just clearly mention the news source and do not mention the timeline. Utilize the following keywords provided in the ⟨*key set*⟩ to construct the question. The correct answer should based on the factual excerpts and is only one word.

⟨*examples*⟩
Your time-sensitive comparison question:

Table 10: Temporal Query Generation Prompting

A multi-hop question is a query requiring multiple inferential leaps or accessing several pieces of information from different locations or sources to arrive at an answer. Considering you have read at least two news articles on ⟨*entity*⟩, construct a multi-hop question that incorporates all the news sources. The source of the news should be stated in the question. Also, ensure that the answer to the question is a single word/entity. Do not answer this question directly. Just give me the question:

Table 11: Null Query Generation Prompting

## B   Appendix B: Dataset Examples

In this appendix, we present an example of each type of multi-hop query included in the MultiHop-RAG dataset. These examples are illustrated in the respective tables: Table 12 for Inference Queries, Table 13 for Comparison Queries, Table 14 for Temporal Queries, and Table 15 for Null Queries. Each query is paired with a ground-truth answer for the evaluation of generation accuracy, while multiple pieces of supporting evidence are included for assessing retrieval performance. Additionally, metadata such as the title, source, and publication time of the news articles are provided as references.

---

**Query:** Which platform is at the center of discussions in articles from Music Business Worldwide, Polygon, and FOX News - Health, concerning the policing of AI-driven voice replication, the debate over "reaction" content, and being the most used app overnight by young people?

**Answer:** YouTube

---

**Evidence List:**

Title: Sony Music's artists aren't involved in YouTube's new voice-cloning AI experiment.
Source: Music Business Worldwide
Published Time: 2023-11-23T18:48:48+00:00
Fact: During this period of discussion, YouTube has made a number of positive announcements regarding the biggest issue for any rightsholder regarding AI-driven voice replication of artists: their ability to police it.

Title: YouTube demonetizes popular content creator SSSniperwolf after doxxing accusations
Source: Polygon
Published Time: 2023-10-25T18:18:06+00:00
Fact: The debate over "reaction" content on YouTube has been brewing for years, but a recent incident between two creators has refueled the urgency of the conversation.

Title: Cell phone shocker as 97% of kids use their device during school hours and beyond, says study Source: FOX News - Health
Published Time: 2023-10-01T09:05:26+00:00
Fact: Overnight phone use was primarily spent engaging with the same media, although YouTube appeared to be the longest-running app because videos were often left playing during the night.

---

Table 12: The example of inference questions

**Query:** Did the Cnbc — World Business News Leader report on Nike's net income and the article from The Age on the 10-year Treasury yield both report a decrease in their respective financial metrics?
**Answer:** Yes

**Evidence List:**
Title: Nike misses revenue expectations for the first time in two years, beats on earnings and gross margin
Source: Cnbc — World Business News Leader
Published Time: 2023-09-28T20:31:00+00:00
Fact: The company's reported net income for the three-month period that ended August 31 was $1.45 billion, or 94 cents per share, compared with $1.47 billion, or 93 cents per share, a year earlier.

Title: ASX set to open higher as Wall Street rebounds; $A rises
Source: The Age
Published Time: 2023-10-04T21:01:01+00:00
Fact: The yield on the 10-year Treasury, which is the centrepiece of the bond market, pulled back from its highest level since 2007, down to 4.73 per cent from 4.80 per cent late on Tuesday.

Table 13: The example of comparison questions

**Query:** Was the performance of the Chicago Bears' defense reported as improved by Yardbarker after Sporting News highlighted a sack by the Bears' defense on Joshua Dobbs during the NFL 'Monday Night Football' game?
**Answer:** Yes

**Evidence List:**
Title: Bears vs. Vikings live score, updates, highlights from NFL 'Monday Night Football' game
Source: Sporting News
Published Time: 2023-11-27T23:32:04+00:00
Fact: The Bears answer right back and sack Dobbs, with Sweat and Brisker in there to take him down.

Title: Hottest seat on each NFC team: Buns burning for these four head coaches
Source: Yardbarker
Published Time: 2023-11-30T22:29:33+00:00
Fact: In his second season as HC, the defense has improved, but positive results are hard to come by behind a lackluster offense ranked 19th in yards (323.2) and 21st in points per game (20.2).

Table 14: The example of time-sensitive questions

**Query:** What is the first letter of the CEO's last name in the news article from Bloomberg on TomTom, and what is the first letter of the city where the company's headquarters is located in the news article from Reuters?
**Answer:** Insufficient information.

Table 15: The example of negative rejection questions

