# OpenReview forum: "MultiHop-RAG: Benchmarking Retrieval-Augmented Generation for Multi-Hop Queries"
_colmweb.org/COLM/2024/Conference — COLM_

### Official Review · Reviewer_rLY3 · 2024-05-05

**Rating:** 3
**Confidence:** 5
**Ethics Flag:** 1

**Summary:**

The paper introduces "MultiHop-RAG," a dataset designed to evaluate retrieval-augmented generation (RAG) systems by focusing on multi-hop queries. The author conducts experiments on the various embedding models, testing their efficacy in evidence retrieval for multi-hop queries and, furthermore, evaluates the ability of off-the-shelf LLMs such as GPT-4 and Llama2-70B,  indicating that direct RAG methods struggle with the multi-hop query demands.

**Reasons To Accept:**

The paper identifies a gap in current RAG systems, specifically their inefficiency in handling multi-hop queries. By focusing on this underexplored area, the paper employs GPT-4 to develop a benchmark and conducts experiments to evaluate both the retrieval and generation components of the RAG systems. However, there remain key issues that prompt reasons for rejection, see next part.

**Reasons To Reject:**

The paper claims to present the first multi-hop RAG (Retrieval-Augmented Generation) dataset. However, this statement overlooks existing datasets such as HotpotQA and 2WikiMultiHop, which also focus on multi-hop reasoning and are cited in the related works section. However, the paper does not adequately differentiate its dataset from these predecessors in the main body of the text. A clarification or rephrasing to accurately position this work within the existing landscape of multi-hop datasets is necessary.

The evaluation methodology appears to inadequately address the core concept of multi-hop reasoning. The described approach primarily involves direct prompting to retrieve relevant documents and generate answers, which seems to neglect advanced multi-hop retrieval and reasoning techniques referenced in prior studies [1,2,3]. More rigorous testing focused on multi-hop reasoning capabilities is essential to substantiate the dataset's utility and effectiveness.

The construction of the dataset raises several concerns. The method involves using GPT-4 to generate answers, which is followed by a quality assurance process. However, the details of this process are vaguely described, such as the proportion of data points filtered out during quality checks and the accuracy of GPT-4 in generating correct answers. For a project aiming to establish a benchmark, the reliance on LLMs for generating dataset content without thorough validation may undermine the credibility and utility of the dataset. Typically, LLMs might be more suitably used for augmenting training sets rather than creating benchmarks intended for evaluating model capabilities.

[1] Yu, W., Zhang, H., Pan, X., Ma, K., Wang, H., & Yu, D. (2023). Chain-of-note: Enhancing robustness in retrieval-augmented language models. arXiv preprint arXiv:2311.09210.

[2] Zhang, J., Zhang, H., Zhang, D., Liu, Y., & Huang, S. (2023). Beam retrieval: General end-to-end retrieval for multi-hop question answering. arXiv preprint arXiv:2308.08973.

[3] Trivedi, H., Balasubramanian, N., Khot, T., & Sabharwal, A. (2022). Interleaving retrieval with chain-of-thought reasoning for knowledge-intensive multi-step questions. arXiv preprint arXiv:2212.10509.

---

> ### Author Rebuttal · Authors · 2024-05-26
>
> Thank you for your insightful comments. We believe there is some misunderstanding of our paper, and we hope to take this opportunity to clarify some important points.
>
> 1. Regarding your comments on “.. overlooks existing datasets such as HotpotQA and 2WikiMultiHop...” we indeed discuss HotpotQA and 2WikiMultiHop in Section 5 “Related Work”. These datasets primarily focus on assessing a model’s reasoning skills, and they do not emphasize the retrieval of evidence from a knowledge base, thereby not tailored for RAG context. Additionally, their primary data sources, Wikipedia, significantly overlap with the training data of most existing LLMs. If we use these sources for benchmarking RAG systems, there is a potential concern that LLM responses might rely on training knowledge rather than reasoning from the retrieved knowledge base.
>
> 2. Regarding your comments on “The evaluation methodology appears to inadequately address the core concept of multi-hop reasoning,” we acknowledge that there are more advanced methods to handle multi-hop queries, yet those methods are benchmarked on widely used public QA corpora. For example, [1] is evaluated on open-domain question-answering benchmarks, which is different from real-world RAG scenarios. Moreover, another outstanding concern is that those benchmark datasets are exposed to LLM pretraining. Multihop-RAG, instead, includes news articles published after September 26, 2023, beyond the knowledge cutoff of some widely-used LLMs, including ChatGPT and LLaMA, as of the time of writing. This selection also helps in teasing out the possibility of the underlying LLM having been exposed to these news articles, thereby simulating a common scenario of RAG where it is used to retrieve from non-public or proprietary datasets. Therefore, we believe Multihop-RAG can serve as a useful resource for benchmarking advanced RAG techniques.
>
> 3. Regarding your comments on "construction of the dataset raises several concerns," to clarify, we do not solely rely on LLMs for dataset construction. Human inspection and manual review by the authors are conducted to ensure data quality. Moreover, we have hired native English-speaking crowd workers to test the human capability of answering the questions. We will provide more detailed information in the revised manuscript.
>
> [1] Yu, W. et.al. (2023). Chain-of-note: Enhancing robustness in retrieval-augmented language models.
>
> Should you have further questions, we would be happy to answer them.

---

> > ### Comment · Reviewer_rLY3 · 2024-06-03
> >
> > I remain unconvinced by the authors' rebuttal. The novelty of this work is still unclear. The response does not adequately address the fundamental difference between this work and existing multi-hop reasoning benchmarks, aside from the retrieval data source. My initial question was not intended to suggest that Wikipedia should be the data source. Rather, I am concerned with the conceptual differences, particularly given that previous literature has also evaluated retrieval performance and generation accuracy of large language models. If the primary contribution of this work is the introduction of a different data source, then the claim of presenting "the first multi-hop RAG (Retrieval-Augmented Generation) dataset" appears tenuous.
> >
> > Regarding my second concern, the authors did not directly address why they did not evaluate using existing multi-hop reasoning baselines. The baselines I mentioned, along with several other widely used multi-hop reasoning methods(such as chain-of-note[1], react[2]), generate follow-up questions and use these questions to retrieve relevant documents before generating a final answer. This approach would seem to be more appropriate as a baseline for evaluating a multi-hop RAG dataset. The choice of data source should not preclude testing against these established methods.
> >
> > In light of these points, I plan to maintain my score.
> >
> > [1] Yu, W. et.al. (2023). Chain-of-note: Enhancing robustness in retrieval-augmented language models.
> >
> > [2] Yao, S., Zhao, J., Yu, D., Du, N., Shafran, I., Narasimhan, K., & Cao, Y. (2022). React: Synergizing reasoning and acting in language models. arXiv preprint arXiv:2210.03629.

---

> > ### Author Response · Authors · 2024-06-03
> >
> > Thank you very much for your reply! We appreciate your academic professionalism and willingness to engage in this discussion. Below, we provide our response to each of your comments in this round of discussion.
> >
> > **Q1. The novelty of this work compared to existing multi-hop reasoning benchmarks**
> >
> > In addition to the retrieval data source, we have designed certain questions to assess the reliability of the target RAG systems, particularly in scenarios involving null queries. This type of question lacks a relevant context in the corpus, necessitating the LLM to identify and reject responses rather than make up a wrong answer. This is another important difference. Since LLM struggles to say "I don’t know" [1] and hallucinations exist in the RAG framework [2], it is important to measure the robustness of RAGs. Thus, our dataset proposes a comprehensive testbed for the RAG framework.
> >
> > **Q2. No evaluation of the advanced framework**
> >
> > We appreciate your insight into the RAG framework and its role in our research. While focusing on dataset construction, we utilized a basic RAG framework to demonstrate the dataset’s potential. We are excited about the more complex frameworks, which could provide further insights, and have acknowledged this as a promising area for future research in the limitation section. And we're still exploring new methods. For your reference, we have tested the performance of advanced RAGs based on generating follow-up questions (query decompose) and using these questions to retrieve relevant documents. We also tested the Hybird search [3]. The baseline in the following table is the basic RAG framework, and the retriever is the best model tested in our work (bge-large-en-v1.5). Here are our results:
> >
> > |    mode   | chunk size | MRR@10 | MAP@10 | Hits@4 |
> > |:---------:|------------|:------:|:------:|:------:|
> > | baseline  | 512        | 0.4975 | 0.2447 | 0.6718 |
> > | hybrid    | 512        | 0.7017 | 0.3682 | 0.8421 |
> > | decompose | 512        |  0.508 | 0.2509 | 0.6794 |
> > | baseline  | 256        | 0.4322 | 0.2017 | 0.6049 |
> > | hybrid    | 256        | 0.6779 | 0.3469 | 0.8018 |
> > | decompose | 256        | 0.4576 | 0.2218 | 0.6222 |
> > | baseline  | 128        | 0.3679 | 0.1728 | 0.4958 |
> > | hybrid    | 128        | 0.5988 | 0.3016 | 0.7135 |
> > | decompose | 128        | 0.3884 | 0.1923 | 0.5188 |
> >
> > It is evident that the advanced framework's performance on our datasets is suboptimal. As a result, we believe Multihop-RAG can be a useful resource for benchmarking advanced RAG technology, and we are available for further discussion if you have any additional questions or need more clarification.
> >
> > [1] Yin, Zhangyue, et al. Do Large Language Models Know What They Don't Know?
> >
> > [2] Niu, Cheng, et al. RAGTruth: A Hallucination Corpus for Developing Trustworthy Retrieval-Augmented Language Models.
> >
> > [3] Gordon V. Cormack, et Reciprocal rank fusion outperforms condorcet and individual rank learning methods.

---

> > > ### Author Response · Authors · 2024-06-03
> > > **Further clarification on the differences between MultiHop-RAG and existing multi-hop datasets**
> > >
> > > Please allow us to further clarify a key difference between MultiHop-RAG and existing datasets such as HotpotQA and 2WikiMultiHop. We acknowledge the existence of multihop QA datasets such as HotpotQA and 2WikiMultiHop, among others. However, these existing datasets are not designed for the RAG task and do not include a retrieval step. The input used in these datasets is already provided with multiple pieces of evidence, and they primarily examine QA models' performance in multi-hop reasoning. Even if one extends HotpotQA to evaluate retrieval performance by combining the HotpotQA input data (Wikipedia text) as a corpus, this extension does not align with real-world RAG tasks. In real-world RAG scenarios, the dataset is usually internal or proprietary corpus, not a corpus used in LLM training such as Wikipedia. This is why we mentioned the Wikipedia data source issue in our previous response.
> > >
> > > Our developed dataset, on the other hand, is specifically designed for the RAG task. It can be used to benchmark multi-evidence retrieval capability (given a question that requires multiple pieces of evidence, how to retrieve relevant pieces of evidence from a database) as well as multi-hop evidence generation capability. Our experiments find that an off-the-shelf RAG solution cannot effectively handle multi-evidence retrieval, which subsequently affects downstream generation performance. We believe the focus on the retrieval aspect sets us apart from existing datasets.

---

> > > > ### Author Response · Authors · 2024-06-07
> > > > **Any further questions?**
> > > >
> > > > Dear Reviewer rLY3,
> > > >
> > > > We appreciate your constructive feedback. As noted in our engagement, we believe there is a misunderstanding regarding the contribution of our work. We want to reiterate that our developed dataset is specifically for RAG tasks, which differs from previous multi-hop QA datasets that primarily focus on the multi-hop reasoning aspect while overlooking the multi-hop retrieval aspects. In addition to this major difference, there are also other outstanding differences, which we do not repeat here in the interest of your time.
> > > >
> > > > If we clear up this misunderstanding, we believe this paper constitutes a valuable resource for benchmarking real-world RAG systems, as most of the human queries in RAG systems involve multi-hop, multi-evidence retrieval, and reasoning, as rightly recognized by other reviewers. We hope we have clarified this point and would appreciate it if you could reconsider your rating. Should you have any further questions, we are very happy to address them.
> > > >
> > > > Best,
> > > > Authors

---

### Official Review · Reviewer_zFdY · 2024-05-09

**Rating:** 7
**Confidence:** 3
**Ethics Flag:** 1

**Summary:**

This paper presents a new dataset to assess multihop RAG, with preliminary performance results by a number of available embeddings and models, with items to test inference, comparison, temporal relations, and a null option to assess hallucination. The dataset is recent (Sept-Dec 2023) so includes data that the tested models were not trained on. The abstract indicates that existing systems perform "unsatisfactorily", and the paper clarifies that conclusion by showing that best performance is 56% or less (table 6).

The paper is well-situated and fairly clear. I think it will be useful.

One shortcoming is in the quality control (step 5). The paper says "we manually review a subset sample", but we are not told how large that sample is, how it was selected, or whether reviewers agreed with each other. A related problem is the complete lack of individual item analysis in the paper. Accuracy per query type is reported, but across those types, were the same items difficult for all systems? And *those* difficult items are the ones that should be manually reviewed, and discussed as examples. A perhaps easier alternative sanity check could be provided with a quick check of human performance on the task. Another shortcoming is the lack of analysis of which features of existing systems best explain the range of performance results. None of these things are listed among the limitations that could be improved with future research.

**Questions To Authors:**

Some informal item analysis must have been done, and could be reported in the quality section (step 5).

**Reasons To Accept:**

The dataset and presentation address an important aspect of language models that needs  improvement.

**Reasons To Reject:**

I think some additional attention should be given to quality control and item analysis. If the dataset is widely used, these will be of interest.

---

> ### Author Rebuttal · Authors · 2024-05-26
>
> Thank you for your constructive comments! Below, we will respond to each of your comments in detail, and we hope can address your concerns satisfactorily.
>
> **Q1. Details of quality control**
>
> Your comment is well taken. Regarding the details of the review process, for each different question type, we randomly sampled 20 samples, which amounted to a total of 60 samples (excluding null queries). We then hired three native English-speaking crowd workers. Provide multiple full documents corresponding to each query and highlight the relevant evidence. We asked the annotators to answer the questions and rate their difficulty from 1 to 5. After keeping the accuracy of generated answers. We also summarized the difficult score distributions in the following table:
>
> | Query Category      | Below 3  |  Above or equal to 3  |
> | ----------- | -----------  | -----------  |
> | Inference Query     | 70%        |     30%      |
> | Comparison Query | 65%        |     35%      |
> | Temporal Query     | 20%         |     80%      |
>
> We can observe that the human assessment of question difficulty aligns with the results shown in Figure 3 of the paper. Temporal and comparison questions are more challenging compared to inference questions.
>
>  **Q2. Lack of analysis of existing poor performance**
>
> We appreciate the insightful feedback. As you rightly point out, the QA performance highlighted in Table 6 is unsatisfactory and warrants further investigation. Our analysis points to two possible reasons:
>
> 1. **Inadequate Retrieval**: The retrieval component in RAG struggles to retrieve relevant evidence effectively based on multi-hop queries from the knowledge base, leading to lower overall response accuracy. This might indicate that existing embedding models fall short in retrieving the most relevant context, suggesting a need for improvements in embedding models.
>
> 2. **Limited Reasoning**: Even when provided with accurate ground-truth evidence, the response accuracy of LLMs, such as Mixtral-8x7B, remains unsatisfactory. This model particularly struggles with comparison and temporal queries (as shown in Figure 3), indicating that the comparison and temporal reasoning capabilities of LLMs are still weak, highlighting areas where further advancements are essential.
>
> By showing the unsatisfactory performance of the Multi-hop RAG and making the dataset publicly available, we hope to spur further research and development in this important field.

---

> > ### Comment · Reviewer_zFdY · 2024-06-04
> >
> > Thanks, these statements clarify the quality control tests you did, though the statement of results here is still too brief for me to understand fully. Were per-item correlations with GPT-4 or any other calculated?

---

> > ### Author Response · Authors · 2024-06-05
> >
> > Thanks for your insights!  Please allow us to further clarify about the quality control tests.
> >
> > First, the table in Q1 shows the difficulty distribution for each question type based on majority voting from crowdsourced annotations. The difficulty ranges from 1 (easiest) to 5 (most difficult). Scores below 3 are considered easy, while 3 and above are considered difficult. Temporal queries, which involve reasoning about time, are the most challenging for humans, followed by comparison queries that require making comparisons. Null queries, which have no answerable questions, are not evaluated and therefore excluded from this data. The results align with the model answer accuracy shown in Figure 3. As expected, the model exhibits the lowest accuracy on comparison and temporal queries, which are identified as the most difficult for humans. In contrast, the model achieves the highest accuracy on inference questions, which appear to be relatively easier based on the difficulty distribution.
> >
> > Second, motivated by your comments, we reviewed the per-item correlations between human evaluators and model-generated responses. In our understanding, the per-item correlation you referred to may be the model's accuracy in answering each question compared to the human-annotated difficulty levels. If we have misunderstood, please correct us. Table 1 shows the accuracy of different models in answering the questions we used in human annotations. GPT-4 remains the best, followed by Google-PaLM. The results in Table 2 specifically analyze the questions the models answered incorrectly. For each model, the table shows the average difficulty scores of the questions it struggled with, as determined by human annotators. The 'nan' values indicate no incorrect questions. Thus, we can find that GPT-4 and PaLM can handle more challenging questions (failed on a high difficulty score), while open-source models struggle with simpler questions, as indicated by the average difficulty scores of their incorrectly answered questions in Table 2. This observation is consistent with the findings in Table 6 of the paper. Combining all the tables, we can clearly see how the difficulty of different types of questions varies for both humans and models. The stronger the model, the higher its accuracy in answering questions, and it can handle more challenging problems.
> >
> > **Table 1: Generation accuracy of LLMs for the sampled questions.**
> >
> > | Model  	| Comparison Query | Inference Query | Temporal Query | Avg. |
> > | ------------ | ----------------- | --------------- | -------------- |-------------- |
> > | GPT-4  				| 40.00%              	| 100%                 | 50.00%           	 | 63.00% |
> > | Google-PaLM   	| 30.00%               	| 100%                 | 50.00%               | 60.00% |
> > | ChatGPT	 		| 0.00%              	    | 100%                 | 40.00%               | 46.67% |
> > | Claude-2.1 		| 10.00%           	    | 80.00%            	| 10.00%               | 33.34% |
> > | Llama-2-70b-chat-hf  	| 0.00%             	 	| 90.00%              | 0.00%             | 30.00%|
> > | Mixtral-8x7B-Instruct  	| 0.00%               	    | 80.00%              |0.00%              | 26.67%|
> >
> > **Table 2: Average human annotated difficult score of the inaccurate questions from different models.**
> >
> > | Model  	| Comparison Query | Inference Query | Temporal Query |
> > | ------------ | ----------------- | --------------- | -------------- |
> > | GPT-4  				| 2.0              	| nan             | 4.6           	 |
> > | Google-PaLM   	| 2.43             	| nan             | 4.6             |
> > | ChatGPT	 		| 2.4              	| nan             | 4.16           |
> > | Claude-2.1 		| 2.55            	| 2.0             	 | 4.55           |
> > | Llama-2-70b-chat-hf  	| 2.4             	 	| 2.0              | 4.4             |
> > | Mixtral-8x7B-Instruct  	| 2.4              	| 2.0              | 4.4             |
> >
> > The findings from our analysis clearly demonstrate that our dataset poses a substantial challenge due to its high difficulty and quality, making it an ideal testbed for RAG systems.
> >
> > We are available for further discussion if you have any more questions or need additional clarification.

---

### Official Review · Reviewer_w5Gn · 2024-05-11

**Rating:** 7
**Confidence:** 5
**Ethics Flag:** 1

**Summary:**

In this paper, the authors propose a multi-hop queries datasets named MultiHop-RAG for evaluating the LLMs' retrieving and reasoning capabilities. The dataset contains various well-designed types including inference, comparison, temporal and null queries. The detailed data statistics also present the integrity and adequacy of the dataset. From the Table 5 and Table 6, we can observe that the various LLMs can not perform the retrieve and reason well in this difficulty scenario, indicating that the dataset is valuable for the future research.

**Reasons To Accept:**

1. The author propose a novelty and effective problem for RAG scenario with LLMs to further investigate.

2. The dataset MultiHop-RAG is constructed comprehensive and rationality, including various query types and categories.

3. The experiments show the dataset is very challenge for various typical LLMs indicating the potential value of the dataset for future research work.

**Reasons To Reject:**

1. The number of query seems a little bit small, the author can collect more in the future.

2. For the Table 6, we can observe that the LLMs can not perform well in this dataset. However, the detailed reasons are not analyzed in the experiment results description.

---

> ### Author Rebuttal · Authors · 2024-05-26
>
> Thank you for your constructive comment! Below, we respond to each of your comments in detail and hope to address your concerns satisfactorily.
>
> **Q1. The number of queries seems a bit small**
>
> Thank you for your observation. Indeed, expanding the dataset could enhance the robustness of our findings. Our current evaluation dataset comprises 2,556 queries, a size determined by two key factors: first, we excluded a significant number of low-quality queries to maintain data integrity; second, we think the current dataset size might be sufficient for preliminary evaluations of retrieval and QA capabilities. For comparison, the widely referenced SciFact dataset (Wadden, D. et al., 2020) contains 1,409 entries. However, we acknowledge the validity of your suggestion and will consider incorporating a larger and more diverse set of queries in future iterations of our work.
>
> **Q2. Further explanation for table 6**
>
> As you rightly point out, the QA performance highlighted in Table 6 is unsatisfactory and warrants further investigation. Our analysis points to two possible reasons:
>
> 1. **Inadequate Retrieval**: The retrieval component in RAG struggles to retrieve relevant evidence effectively based on multi-hop queries from the knowledge base, leading to lower overall response accuracy. This might indicate that existing embedding models fall short in retrieving the most relevant context, suggesting a need for improvements in embedding models.
>
> 2. **Limited Reasoning**: Even when provided with accurate ground-truth evidence, the response accuracy of LLMs, such as Mixtral-8x7B, remains unsatisfactory. This model particularly struggles with comparison and temporal queries (as shown in Figure 3), indicating that the comparison and temporal reasoning capabilities of LLMs are still weak, highlighting areas where further advancements are essential.
>
> Per your suggestion, we will include these explanations in the main text. Thanks!
>
> We appreciate your thorough review and remain available for further discussion if you have any additional questions or require more clarification.

---

### Official Review · Reviewer_AfhS · 2024-05-13

**Rating:** 8
**Confidence:** 5
**Ethics Flag:** 1

**Summary:**

The paper introduces a novel dataset named MultiHop-RAG, specifically designed to address the inadequacies of existing Retrieval-Augmented Generation (RAG) systems in managing multi-hop queries. The authors establish that current RAG setups fall short in accurately handling these complex queries and highlight the absence of a dedicated benchmarking framework to assess RAG performance on multi-hop inquiries. The developed dataset, MultiHop-RAG, comprises a knowledge base built from English news articles, numerous multi-hop queries, their correct answers, and identifiable supporting evidence.

**Reasons To Accept:**

The development of MultiHop-RAG addresses a glaring gap in the research on RAG systems which is their inadequacy in handling multi-hop queries. This dataset provides a foundational step towards enhancing RAG performance and extending LLM applicability. By making the dataset and benchmarking code publicly available, the paper offers valuable resources for the wider research community. This openness facilitates replication and further research, potentially accelerating progress in the area.

**Reasons To Reject:**

While the paper introduces a new dataset and conducts experiments, it may lack a clear comparison to existing datasets. A comparative analysis demonstrating the unique challenges and benefits of MultiHop-RAG over previous datasets could strengthen the case for its necessity and utility.
In addition, the use of an English news article dataset as the sole source for the knowledge base might introduce bias or limit the scope of queries the dataset can effectively support. The paper could benefit from a discussion on these limitations and potential mitigation strategies.

---

> ### Author Rebuttal · Authors · 2024-05-26
>
> Thank you for your constructive comments. In the following sections, we will provide detailed responses to each of your comments, aiming to address your concerns satisfactorily.
>
> **Q1. Lack of comparison to existing datasets**
>
> We completely agree with you that a comparative analysis demonstrating the unique challenges and benefits of MultiHop-RAG over previous datasets could strengthen the case for its necessity and utility. We have discussed related datasets in the introduction and related work sections of our paper. To clarify and emphasize our dataset's unique position, we list the limitations of related datasets as follows:
>
> * Related Datasets:
> 	1. RAG Evaluation Datasets: RGB (Chen et al., 2023) ,RECALL (Liu et al., 2023)
> 		*  These datasets focus on single-hop questions and primarily measure the accuracy of final answers, not retrieval robustness.
>
> 	2. Fact Extraction datasets: FEVER (Thorne et al., 2018), SciFact (Wadden et al., 2020), HoVer (Jiang et al., 2020), HoVer (Jiang et al., 2020)
> 		*  These datasets neglecting broader QA performance and RAG framework robustness.
>
> 	3.  Multi-document QA datasets: HotpotQA (Yang et al., 2018), MultiRC (Khashabi et al., 2018), 2WikiMultiHopQA  (Ho et al., 2020).
> 		* Existing datasets predominantly utilize Wikipedia as their main data source, lack emphasis on the robustness of the RAG framework, and do not focus adequately on evidence retrieval capabilities.
>
> * MultiHop-RAG (this work): This work uses multi-hop questions from multiple documents to test the robustness of the RAG framework, ideal for evaluating its performance in large-scale corpus retrieval and complex question-answering scenarios. Moreover, the dataset includes news articles published from September 26, 2023, to December 26, 2023, beyond the knowledge cutoff of some widely-used LLMs, including ChatGPT and LLaMA, as of the time of writing. This selection also helps in teasing out the possibility of the underlying LLM having been exposed to these news articles, thereby simulating a common scenario of RAG where it is used to retrieve from non-public or proprietary datasets.
>
> **Q2. English news articles as the sole source.**
>
> We acknowledge that relying exclusively on English news articles as a data source can introduce bias and potentially limit the scope of our queries. To address this limitation, we will discuss this in the limitations section of our paper and consider diversifying our sources in future research.

---

### Decision · Program_Chairs · 2024-07-10

**Decision:**

Accept

**Comment:**

The authors introduce a new dataset, MultiHop-RAG, that uses GPT-4 to construct question-answer pairs based on synthesizing multiple news articles. Retrieving and integrating information from multiple sources is an important and topical challenge for present RAG systems, so the reviewers generally found this to be a timely contribution. However, concerns were raised regarding a lack of details in some places and especially the comparisons to prior work. While MultiHop-RAG is a useful contribution (e.g., it is not Wikipedia-based, so it helps to diversify our current suite of multi-hop QA benchmarks), the authors' claim that it is one of the first RAG datasets to focus on multi-hop queries seems overly strong. I encourage the authors to contextualize MultiHop-RAG appropriately and to incorporate their responses to the reviewers into the next version.

[comments from the PCs] Please refine your claim presentation following the AC recommendation.